# Base Metals Extraction from Printed Circuit Boards by Pressure Acid Leaching

**Guadalupe Martinez-Ballesteros [1],\*, Jesus Leobardo Valenzuela-Garcia [1],\*[ID], Agustin Gomez-Alvarez [1], Martin Antonio Encinas-Romero [1], Flerida Adriana Mejia-Zamudio [1] and Aaron de Jesus Rosas-Durazo [2]**

[1]   Department of Chemical Engineering and Metallurgy, University of Sonora, Hermosillo 83000, Mexico
[2]   Department of Biomedical, Technological Institute of Hermosillo, Hermosillo 83100, Mexico
\*   Correspondence: guadalupe.martinezballesteros@unison.mx (G.M.-B.);
     jesusleobardo.valenzuela@unison.mx (J.L.V.-G.)

**Abstract:** Printed circuit boards (PCBs) are a valuable source of raw materials for metal recycling considering that base metal concentration analyses have confirmed that PCB powders are multi-metallic in nature and contain high concentrations of Cu, Zn, Ni, and Fe. Given that minerals are not renewable resources, these metals can be recycled through hydrometallurgical processes. In this study, we determined that 2 M sulfuric acid, 0.55 MPa oxygen pressure, and a temperature of 90 °C represent the optimal conditions for leaching of Cu, Zn, and Ni of PCBs, obtaining the highest observed values of recovery of greater than 90% for Zn and 98% for Cu and Ni. The characterization of PCBs by SEM–EDS analyses showed that plates mainly consist of Cu, Ni, and Zn. PCBs can be seen as a potential secondary resource for the recovery of copper, nickel, and zinc. The best potential and pH conditions for the extraction of Cu, Zn, and Ni were also determined on the basis of thermodynamic diagrams.

**Keywords:** pressure leaching; printed circuit boards; base metals; recycling

## 1. Introduction

The production of electrical and electronic equipment (EEE) is increasing rapidly due to the revolution of information technology; accordingly, the waste deriving from electrical and electronic equipment (WEEE) has also increased [1]. WEEE comes from private homes and as a result of professional use. Improvements in the processing capacity of computers have resulted in a decrease in their average lifespan. Therefore, the amount of WEEE each year has increased faster than other type of waste [2]. In Mexico, there is an estimated generation of 370,724 tons/year of electronic waste, of which 10% is recycled, 40% remains stored at home or in warehouses, and 50% ends up in landfills.

WEEE contains up to 61% metals; hence, it represents an important source of base and precious metals with high economic potential [3]. For example, the printed circuit board (PCB) of a PC can contain up to 20% Cu and 250 g/ton Au [4]. These residues are a potential source of metal recycling since minerals are considered to be non-renewable resources, so recovering these metals could reduce mine exploitation.

Hydrometallurgical and pyrometallurgical processes can be used for metal recovery. In general, pyrometallurgical processes are more expensive due to the high temperatures used. In contrast, hydrometallurgical processes consist of a first stage, in which metals are extracted from the chemical leaching solution in an acid or alkaline medium, followed by a second step that consists of purification of the solution. Methods of precipitation, cementation, adsorption, ion exchange, and solvent extraction are currently used to concentrate and purify solutions from leaching [5–7]. Pyrometallurgical processes are currently the most widely used method for metal recycling. These activities are still carried out employing artisanal processes that are not regulated and where there is no control of the contaminants that result when this type of waste is subjected to high temperatures. Since WEEE

contains a wide range of metals, it also contains organic compounds, which are generally toxic and potentially bioaccumulate, such as brominated flame retardants, polychlorinated dibenzo-p-dioxins, dibenzofurans, polybrominated diphenyl ethers, chlorinated dioxins, and polycyclic aromatics, which produce highly carcinogenic gases when burned [8,9].

The composition of WEEE is much more complex than naturally mined materials; this permits the recovery of valuable metals in more enhanced stages of refinement via the efficient and environmentally sound processing of WEEE [5]. Consequently, a viable method for recycling this waste could be through hydrometallurgical processes since these do not generate as many gases and can separate metals of interest in a more selective way.

In recent years, there has been great interest in recovering metals from PCBs through a leaching process. Mecucci and Scott studied Cu, Pd, and Sn recovery from PCBs by leaching with $HNO_3$ [10]. Castro and Martins leached dust from cards using 2.18 N $H_2SO_4$, 2.18 N $H_2SO_4$ + 3.0 N HCl, 3.0 N HCl, and 3.0 N HCl + 1.0 N $HNO_3$ to extract Cu and Sn in these residues [11]. Yang et al. investigated PCB treatment with a 2 M $HNO_3$ solution for 3 h; Sn could be separated as stannic acid and Pb as a $PbSO_4$ precipitate [12]. Kumar et al. designed two sets of experiments to optimize the operating parameters for the extraction of Cu from printed circuit boards using $HNO_3$ and $H_2SO_4$ in the presence of $H_2O_2$ [13]. Yang et al. used a spent Sn solution containing $HNO_3$, generated through the production of PCBs, which was used for PCB leaching for 2 h at temperature [14]. Kamberovic et al. leached PCBs using a solution of 2 M $H_2SO_4$ and $H_2O_2$ (30 vol %) as an oxidizing agent, with 10% in solids and a temperature of 70 °C [15]. Kavousi et al. used a 2 M solution of fluoroboric acid ($HBF_4$) as a leaching reagent and 0.6 M $H_2O_2$ as an oxidant agent for PCB leaching for 3 h at 75 °C and S/L ratio = 1:15 [16]. Khayyam et al. studied the recovery of metals from printed circuit boards using $HNO_3$ as a leaching reagent and without adding any type of additive or oxidizing agent [17].

In this study, we investigated the leaching of base metals (Cu, Zn, and Ni) from PCBs. Due to the complexity of the material and the presence of metals in native form and as alloys, an oxidative leaching process is required for the effective extraction of metals of interest. $H_2SO_4$ was used as the leachate and $O_2$ as an oxidant. It was decided to use pressure leaching because less leaching time is needed, and a higher percentage of solids can be used compared to other investigations. Sulfuric acid was used as a leaching reagent since it is commonly used in basic metal leaching because it can dissolve these metals, and it is low-cost. The thermodynamic characteristics of metal leaching and the effects of pressure and temperature on the extraction of Cu, Zn, and Ni are discussed here. The material characteristics were subjected to SEM–EDS and chemical analyses.

## 2. Materials and Methods

### 2.1. Materials

Printed circuit boards required to carry out the experimental work were provided by Retroworks of México S.A. de C.V. located in Fronteras, Sonora, Mexico. The concentration of metals in printed circuit boards (PCBs) was determined by atomic absorption spectroscopy (Perkin Elmer model AAnalyst 400 atomic absorption equipment, PerkinElmer Inc, Waltham, MA, USA); the metal concentrations determined in the PCBs are shown in Table 1. The pH and redox potential in solution were analyzed after leaching using a Thermo Scientific Orion Star A111 pH benchtop meter (Thermo Fisher Scientific Inc, Waltham, MA, USA); this allowed us to construct a thermodynamic diagram of Eh–pH and identify and characterize the species in solution. The solid residues were analyzed to determine the characteristics of the material after leaching by scanning electron microscopy and energy-dispersive spectroscopy (SEM–EDS) using a Thermo Scientific Phenom Pro-X (Thermo Fisher Scientific Inc, Waltham, MA, USA). The Pourbaix diagrams were calculated using the HSC 6.0 program for each metal–electrolyte system under corresponding conditions. The chemicals ($H_2SO_4$(aq) and $O_2$(g)) were of analytical grade (≥95%).

**Table 1.** Chemical composition of printed circuit boards.

| Metal | Ag | Au | Pt | Pd | Cu | Zn | Ni | Fe |
|---|---|---|---|---|---|---|---|---|
| Content | 170 (g/t) | 220 (g/t) | 2 (g/t) | 1.2 (g/t) | 13.14% | 0.02% | 4% | 4.9% |

### 2.2. Methods

To avoid interference in the leaching of metals, the PCBs were first subjected to a 24 h treatment with sodium hydroxide (10 M) for the removal of epoxide material; then, for the removal of sodium hydroxide, the PCBs were rinsed with plenty of water [18]. Later, they were cut into pieces of approximately 2 cm × 2 cm for further pulverization at a particle size of −177 μm [19]. Extraction of the metals of interest for leaching was carried out with sulfuric acid ($H_2SO_4$) as a leaching agent, present at 20% in solids, and tap water. We added the pulp into the Titanium PARR Pressure Reactor (capacity of 1 L is equipped with a heating jacket which is controlled by a PARR controller model 4848, which also controls the agitation and pressure of oxygen inside the reactor; it is also equipped with cooling water to efficiently control the temperature inside the reactor) and processed it by stirring at 600 rpm for 4 h, varying the $H_2SO_4$ concentration (1 and 2 M), pressure (0.34, 0.44, and 0.55 MPa), and temperature (60, 75, and 90 °C). After the established contact time, the solution was filtered to separate the solid from the liquid. The pregnant solutions were analyzed using the atomic absorption spectroscopy technique (AAS) to determine their metallic concentration, while solid residues were analyzed to determine the characteristics of the material by SEM–EDS. The flowsheet of Cu, Zn, and Ni leaching from PCBs is given in Figure 1.

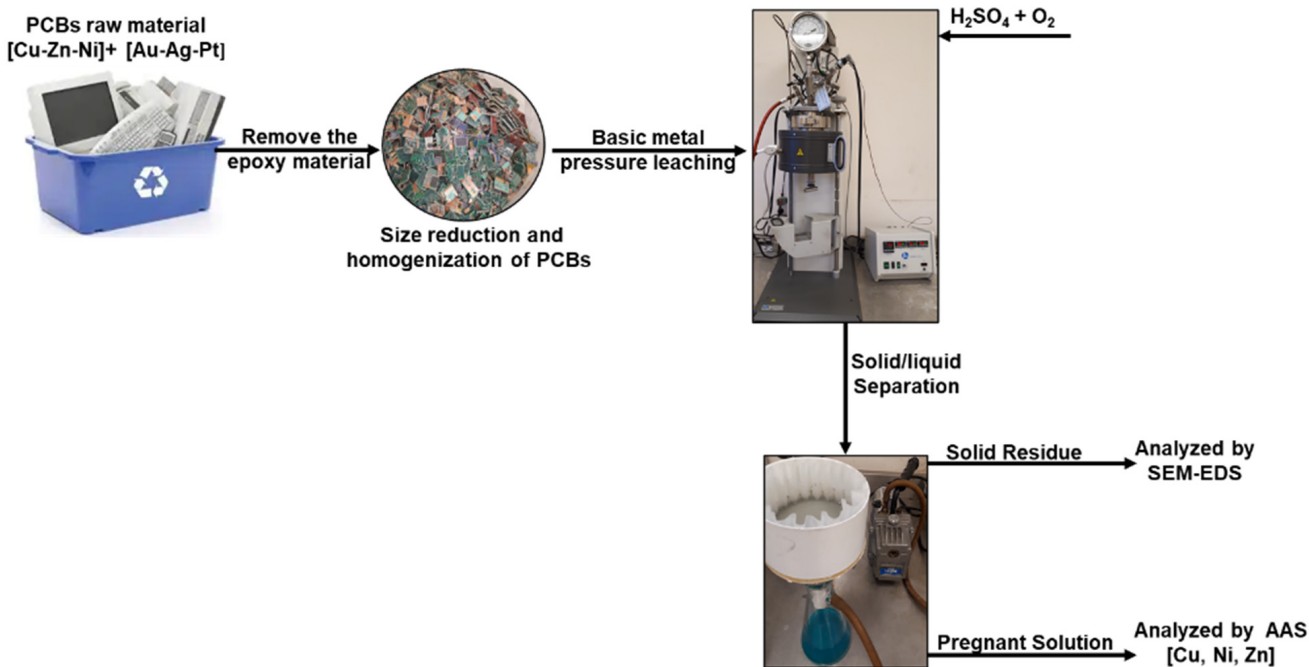

**Figure 1.** Flowsheet of Cu, Zn, and Ni leaching from PCBs.

### 3. Results

#### 3.1. Pressure Leaching

##### 3.1.1. Copper

The effect of pressure and temperature on copper leaching is shown in Figure 2. The extraction of Cu (%) as a function of pressure (MPa) at different temperatures (°C) is shown in Figure 2A at a concentration of 1 M $H_2SO_4$. It can be observed that at a higher temperature, the extraction percentage decreases, where an extraction of 45% of Cu was achieved by using 60 °C and 0.55 MPa. Using a concentration of 2 M $H_2SO_4$, can be seen

that as the temperature rise the percentage of extraction increases; a Cu extraction efficiency of more than 98% was obtained at a temperature of 90 °C and P = 0.55 MPa, as shown in Figure 2B. The reason for the difference in the percentage of extraction at each concentration is because at 1 M $H_2SO_4$, there are no pH and Eh conditions for copper to be present as $CuSO_4$; however, at 2 M $H_2SO_4$, these conditions are present. In Figure 2, it can be seen that the greater the pressure, the greater the extraction of copper; this is because the presence of oxygen is necessary to ensure that the metallic copper present in the PCBs goes into the solution as $CuSO_4$, according to the reaction (1).

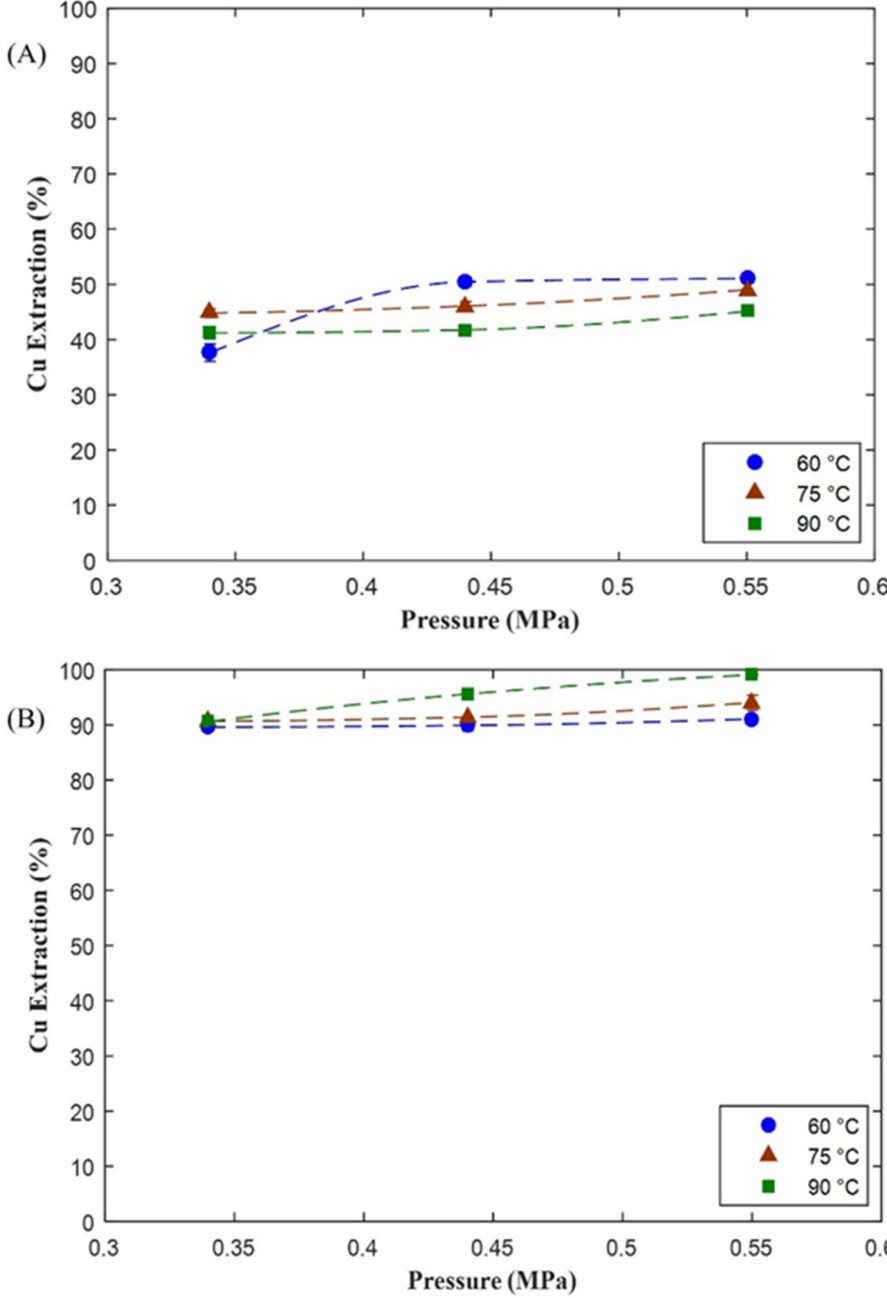

**Figure 2.** Cu extraction (%) as a function of pressure (MPa) at different temperatures (°C): (**A**) at 1 M $H_2SO_4$; (**B**) at 2 M $H_2SO_4$ (stirring at 600 rpm, for 4 h, 20% solids, and a particle size of −177 μm).

3.1.2. Zinc

The effect of pressure and temperature on zinc leaching is shown in Figure 3. It can be seen that as the temperature increases, the percentage of extraction increases in the two

molarities of sulfuric acid (1 and 2 M) used during this investigation. The extraction of Zn (%) is shown as a function of pressure (MPa) at different temperatures (°C). Extraction above 75% was obtained by using a temperature of 90 °C, 0.55 MPa, and 1 M $H_2SO_4$, as shown in Figure 3A. An extraction efficiency above 90% was obtained at a concentration of 2 M $H_2SO_4$, as shown in Figure 3B. The difference in the percentage of extraction in each concentration is that at 2 M $H_2SO_4$, $ZnSO_4$ presents greater stability. In Figure 3, it can be seen that the greater the pressure, the greater the extraction of zinc; this is because the presence of oxygen is necessary to ensure that the metallic zinc present in the PCBs goes into the solution as $ZnSO_4$, according to the reaction (2).

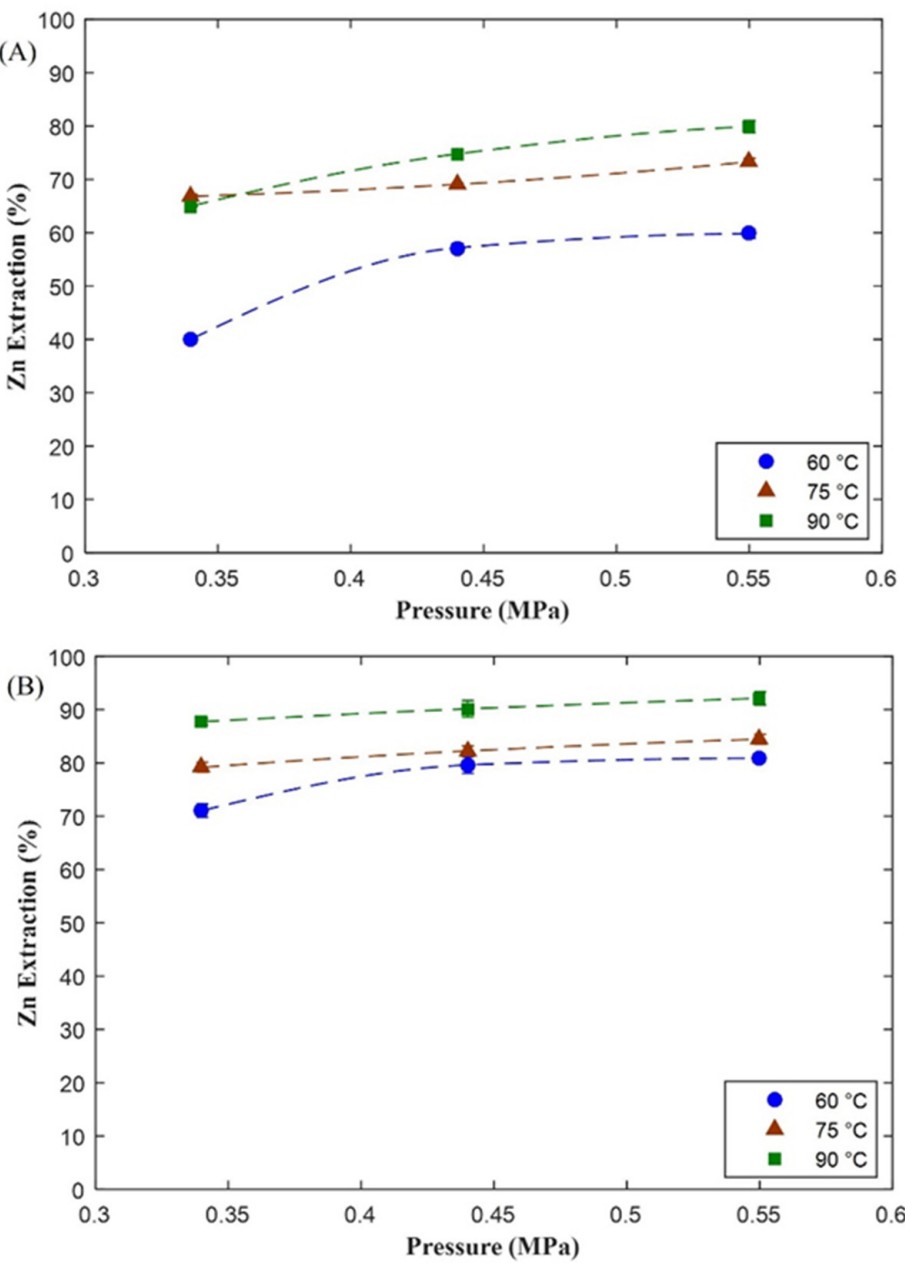

**Figure 3.** Zn extraction (%) as a function of pressure (MPa) at different temperatures (°C): (**A**) at 1 M $H_2SO_4$; (**B**) at 2 M $H_2SO_4$ (stirring at 600 rpm, for 4 h, 20% solids and a particle size of $-177$ μm).

3.1.3. Nickel

The effects of pressure and temperature on nickel leaching are shown in Figure 4. It is observed that with a higher temperature, a higher percentage of extraction is obtained.

The extraction of Ni (%) is shown as a function of pressure (MPa) at different temperatures (°C). The best extractions were obtained by using 90 °C and 0.55 MPa, above 90% at a concentration of 1 M $H_2SO_4$ (Figure 4A) and greater than 98% at a concentration of 2 M $H_2SO_4$ (Figure 4B). The difference in the percentage of extraction in each concentration is that at 2 M $H_2SO_4$, $NiSO_4$ presents greater stability. In Figure 4, it can be seen that the greater the pressure, the greater the extraction of nickel; this is because the presence of oxygen is necessary to ensure that the metallic nickel present in the PCBs goes into the solution as $NiSO_4$, according to the reaction (3).

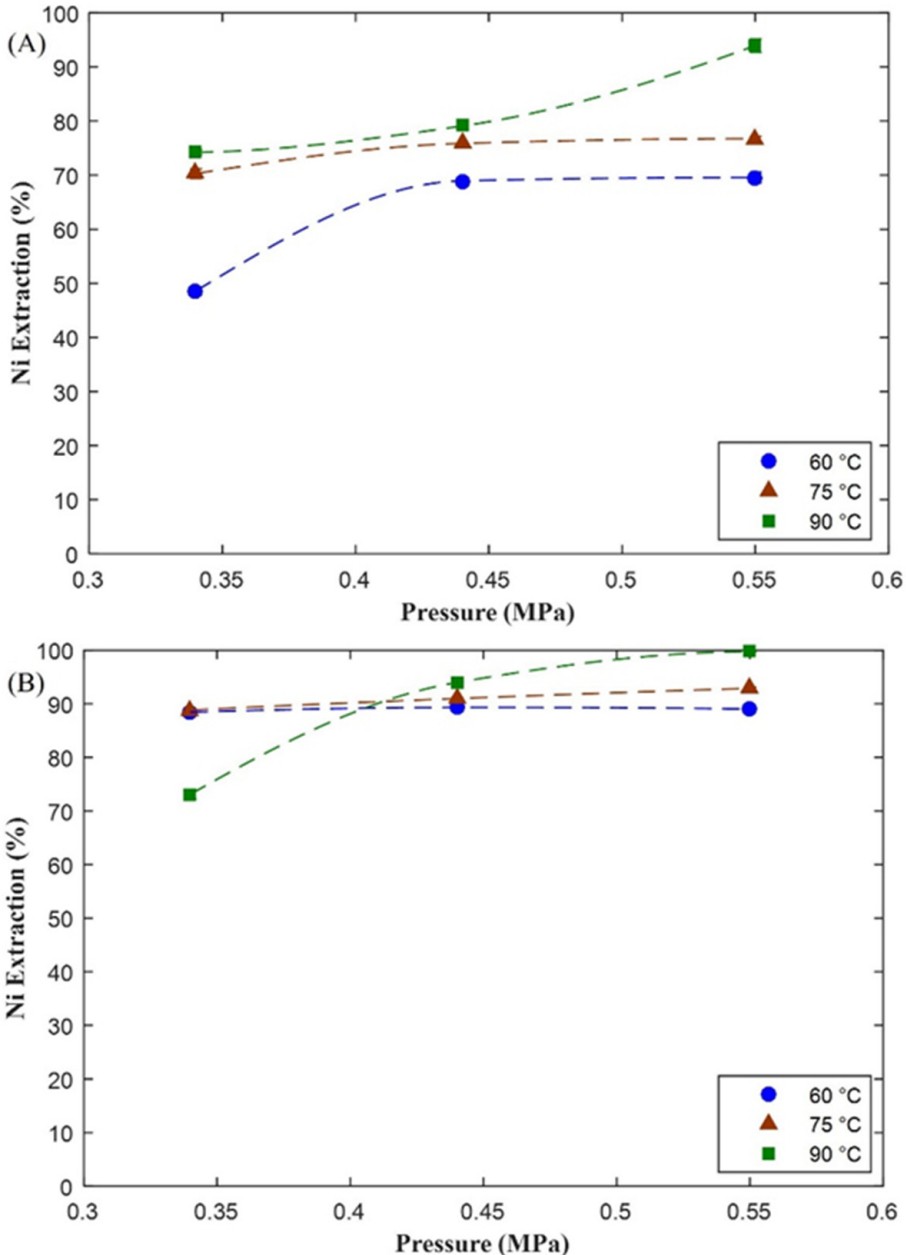

**Figure 4.** Ni extraction (%) as a function of pressure (MPa) at different temperatures (°C): (**A**) at 1 M $H_2SO_4$; (**B**) at 2 M $H_2SO_4$ (stirring at 600 rpm, for 4 h, 20% solids and a particle size of −177 μm).

### 3.2. Thermodynamic Analysis

The reactions that occur during the leaching of base metals (Cu, Zn, and Ni) using sulfuric acid as a lixiviant are set out in Equations (1)–(3), considering that they are in a metallic form in PCBs [20]:

$$Cu^\circ_{(s)} + \tfrac{1}{2}O_{2(g)} + H_2SO_{4(aq)} \to Cu^{2+}_{(aq)} + SO^{2-}_{4(aq)} + H_2O$$
$$\Delta G_{90^\circ C} = -1120.6 \tfrac{kJ}{mol} \tag{1}$$

$$Zn^\circ_{(s)} + \tfrac{1}{2}O_{2(g)} + H_2SO_{4(aq)} \to Zn^{2+}_{(aq)} + SO^{2-}_{4(aq)} + H_2O$$
$$\Delta G_{90^\circ C} = -1201.77 \tfrac{kJ}{mol} \tag{2}$$

$$Ni^\circ_{(s)} + \tfrac{1}{2}O_{2(g)} + H_2SO_{4(aq)} \to Ni^{2+}_{(aq)} + SO^{2-}_{4(aq)} + H_2O$$
$$\Delta G_{90^\circ C} = -1078.67 \tfrac{kJ}{mol} \tag{3}$$

Pourbaix diagrams were constructed for each metal–electrolyte system using HSC 6.0 software, 90 °C, 0.55 MPa, and the corresponding metal concentration. The two points indicate species present at the different molarities of $H_2SO_4$ used in the experiments according to the potential redox (Eh) and pH (1 M (Eh = 0.42 v, pH = 3.5) and 2 M (Eh = 0.56 v, pH = 0.2)). The Eh–pH diagram for the Cu–S–$H_2O$ system is shown in Figure 5, where we can see that Cu species exist in solution as copper oxide ($Cu_2O$) when a solution of 1 M $H_2SO_4$ is used. On the other hand, when a solution of 2 M $H_2SO_4$ and acid medium (pH < 1) is used, aqueous copper sulfate ($CuSO_4$) is present, as seen in the Pourbaix diagram (Figure 5). Therefore, there is higher extraction of Cu using 2 M rather than 1 M of $H_2SO_4$, as shown in Figure 2.

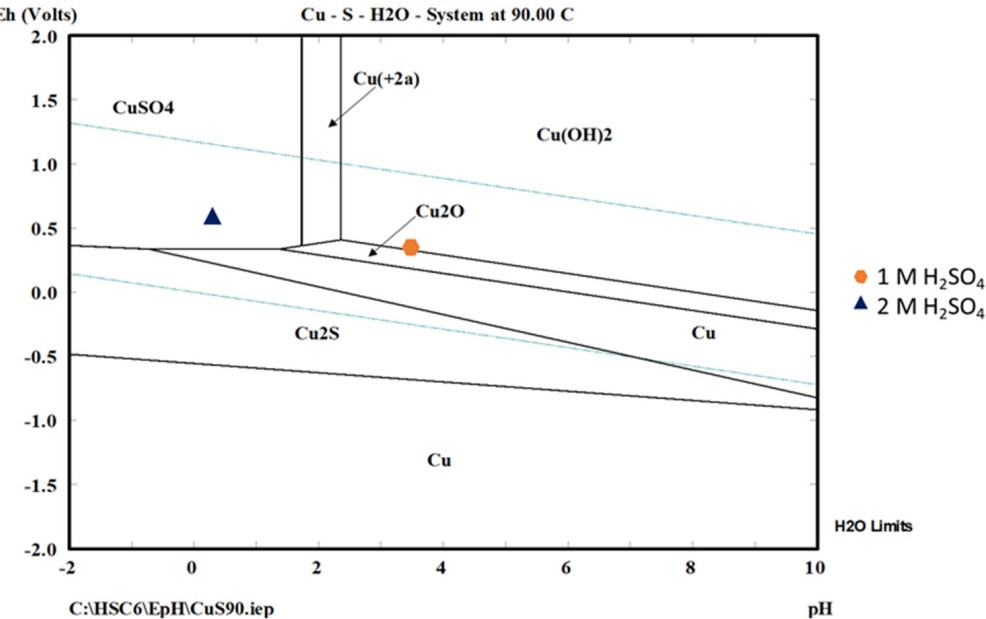

**Figure 5.** Pourbaix diagram for the Cu–S–$H_2O$ system at [Cu] = 0.95 M, 90 °C, and 0.55 MPa.

The Pourbaix diagram for the Zn–S–$H_2O$ system is shown in Figure 6, where Zn species exist in solution as zinc sulfate ($ZnSO_4$) for both tested concentrations of $H_2SO_4$; there is a greater area of stability at 2 M $H_2SO_4$, as shown in Figure 3. Therefore, the extraction of Zn is more significant at this concentration.

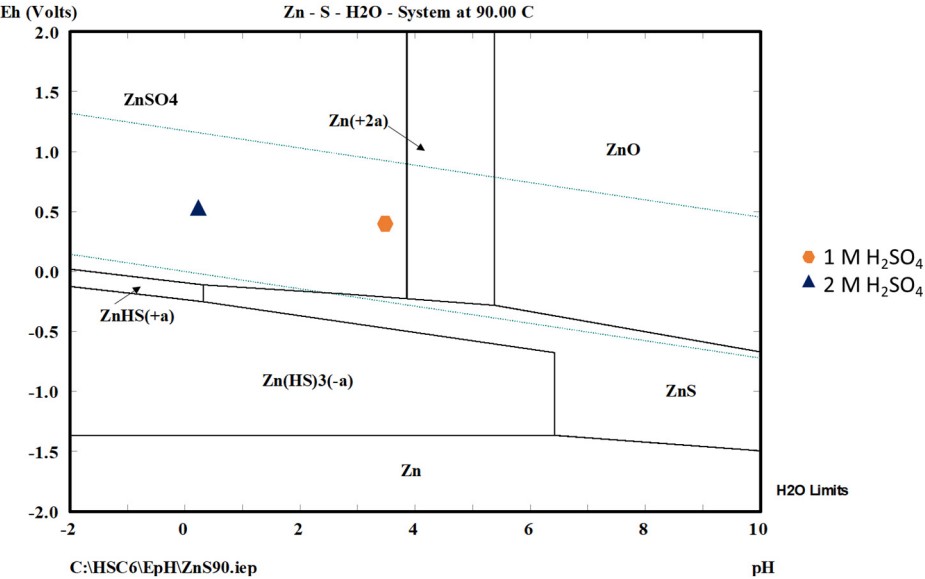

**Figure 6.** Pourbaix diagram for the Zn–S–H$_2$O system at [Zn] = 0.005 M, 90 °C, and 0.55 MPa.

The Pourbaix diagram for the Ni–S–H$_2$O system is shown in Figure 7, where Ni species exist in solution as nickel sulfate (NiSO$_4$) for both concentrations of H$_2$SO$_4$; there is a greater area of stability at 2 M H$_2$SO$_4$, as shown in Figure 4. Therefore, the extraction of Ni is more significant at this concentration.

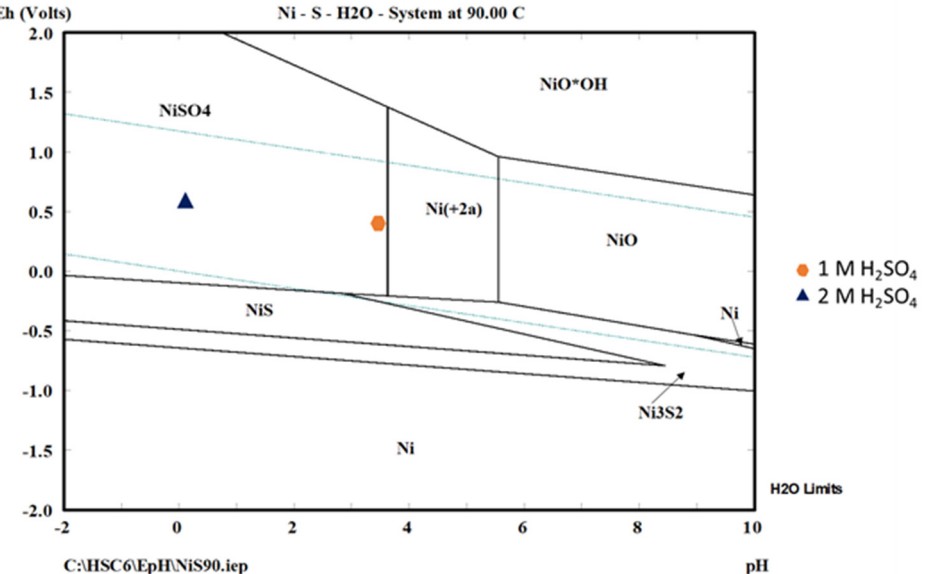

**Figure 7.** Pourbaix diagram for the Ni–S–H$_2$O system at [Ni] = 0.017 M, 90 °C, and 0.55 MPa.

### 3.3. Study of PCB Material from the Leaching Tests by SEM–EDS

Micrographs and microanalyses of PCBs material before leaching with H$_2$SO$_4$ for the recovery of Cu, Zn, and Ni are shown in Figure 8, in which the material morphology demonstrates high diversity. According to EDS, the primary metals that make up the PCBs are copper, nickel, aluminum, and gold. It is worth mentioning that the base composition of PCBs has multiple elements; consequently, a large majority of elements can be found in their composition. For example, this type of equipment uses its plastic covering and brominated flame-retardant PCBs (BFR) to avoid flammability [21]. This explains the presence of Br in the EDS analysis.

The results of micrography and microanalysis of the sample leached with 2 M $H_2SO_4$ at 0.55 MPa and 90 °C are shown in Figure 9. Under these conditions, the highest extraction of base metals contained in the sample was achieved. The analysis of morphology shows that the structure is composed of plates, and unlike the PCB material (Figure 8), this does not present spheres and cylinders. The results obtained from the EDS show mainly the presence of silver and gold. Other identified metals, such as tin, are present in small amounts.

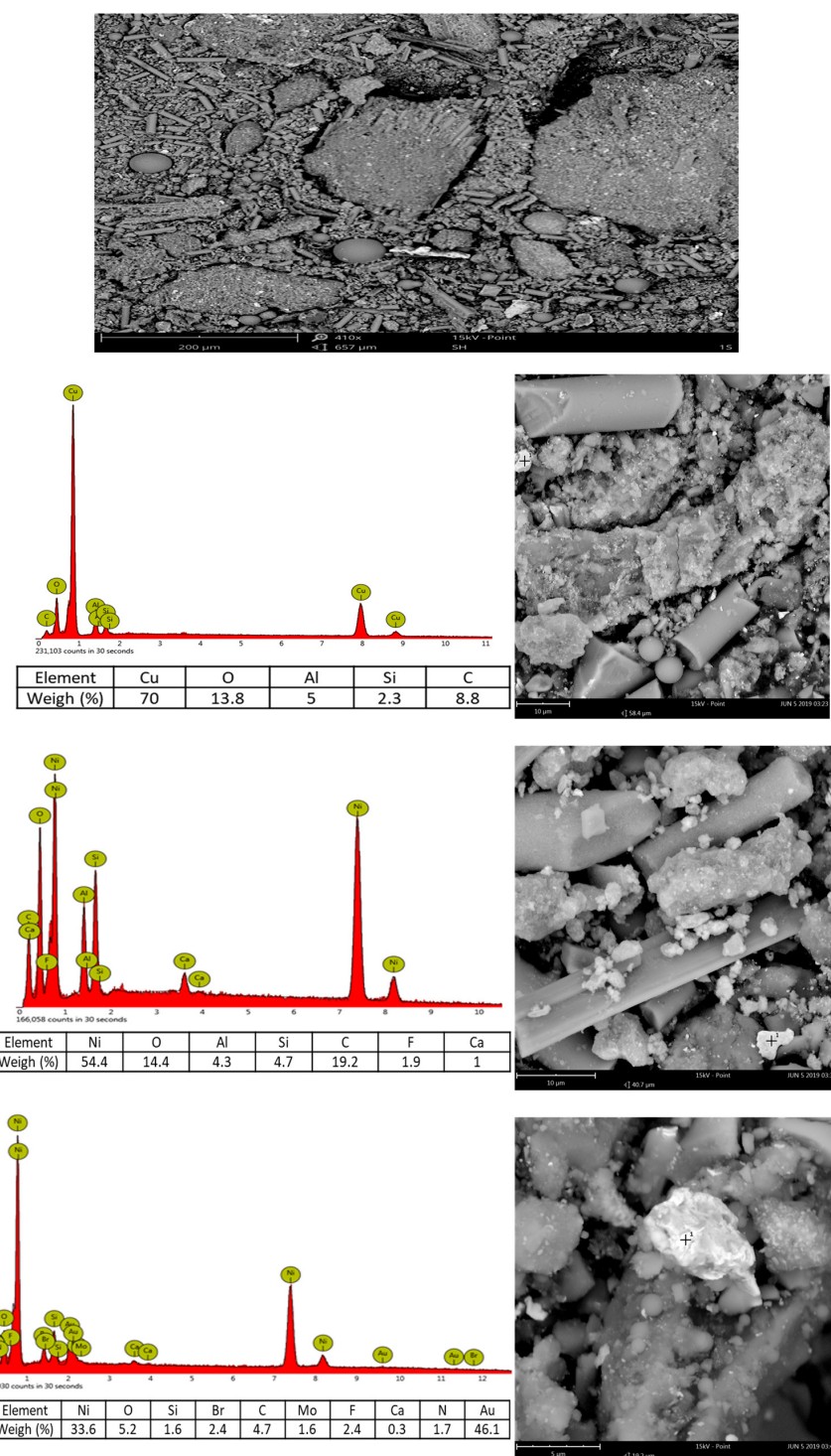

| Element | Cu | O | Al | Si | C | | | | |
|---|---|---|---|---|---|---|---|---|---|
| Weigh (%) | 70 | 13.8 | 5 | 2.3 | 8.8 | | | | |

| Element | Ni | O | Al | Si | C | F | Ca | | |
|---|---|---|---|---|---|---|---|---|---|
| Weigh (%) | 54.4 | 14.4 | 4.3 | 4.7 | 19.2 | 1.9 | 1 | | |

| Element | Ni | O | Si | Br | C | Mo | F | Ca | N | Au |
|---|---|---|---|---|---|---|---|---|---|---|
| Weigh (%) | 33.6 | 5.2 | 1.6 | 2.4 | 4.7 | 1.6 | 2.4 | 0.3 | 1.7 | 46.1 |

**Figure 8.** Scanning electron microscopy (SEM–EDS) images, micrographs, and microanalysis of the PCB sample before leaching.



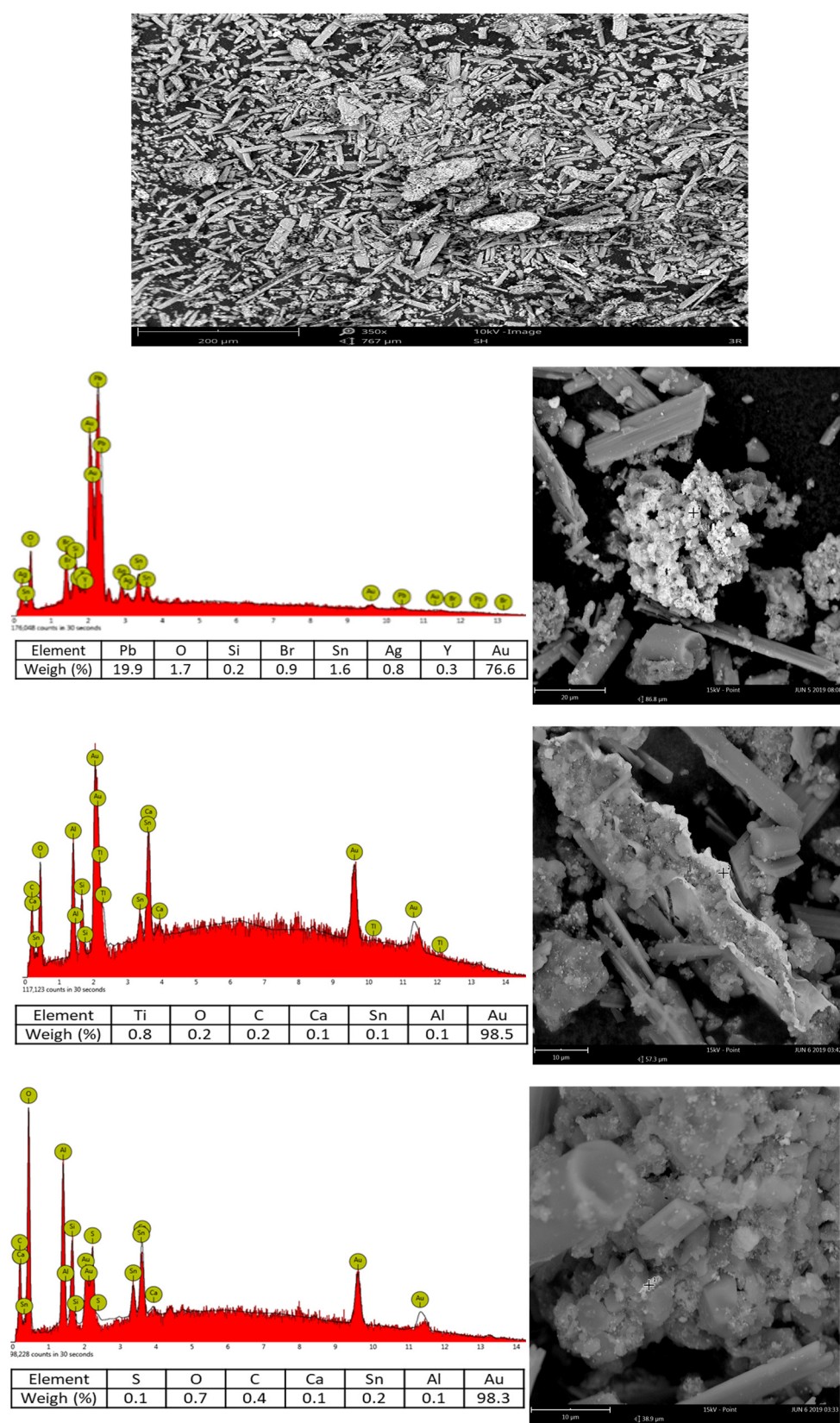

**Figure 9.** Scanning electron microscopy (SEM–EDS) images, a micrograph, and microanalysis of the PCB sample leached with 2 M H$_2$SO$_4$ at 0.55 MPa and 90 °C.

## 4. Discussion

Based on the micrographs of the raw material, a diverse morphology was observed, and the microanalysis shows that it contains several metals: Cu, Ni, Al, and Au. Once the material has been leached with sulfuric acid, a morphology characterized by plates containing Ag and Au can be seen. This material no longer presents spheres and cylinders, and Cu and Ni are not observed.

The concentration of $H_2SO_4$ significantly affects the leaching of copper. The results indicate higher recovery at a concentration of 2 M, since this metal precipitates at 1 M $H_2SO_4$ based on the Pourbaix diagram for the Cu–S–$H_2O$ system. Additionally, there is co-precipitation of other metals. On the other hand, at a concentration of 2 M $H_2SO_4$, Zn and Ni remain as aqueous sulfates. Therefore, according to the Eh–pH diagrams for the Zn–S–$H_2O$ and Ni–S–$H_2O$ systems, the extraction percentages of Zn and Ni are higher.

## 5. Conclusions

Of the conditions tested for Cu, Zn, and Ni leaching in our study, 0.55 MPa pressure, 90 °C temperature, and 2 M $H_2SO_4$ were found to be the optimal conditions leading to the highest metal recovery. Under these conditions, more than 98% of Cu and Ni and 90% of Zn were extracted, as confirmed by the Eh–pH diagrams, mainly because they lay the foundation for thermodynamically favorable conditions for metal leaching. This study demonstrates the feasibility of recovering base metals from PCBs at moderate temperatures and pressures through leaching with sulfuric acid.

**Author Contributions:** Conceptualization, G.M.-B. and J.L.V.-G.; formal analysis, G.M.-B. and F.A.M.-Z.; investigation, G.M.-B., J.L.V.-G., and A.G.-A.; methodology, G.M.-B., J.L.V.-G., A.G.-A., and A.d.J.R.-D.; project administration, J.L.V.-G.; resources, M.A.E.-R. and J.L.V.-G.; data curation, A.G.-A. and F.A.M.-Z.; formal analysis, G.M.-B. and J.L.V.-G.; supervision, J.L.V.-G. and A.G.-A.; validation, G.M.-B. and M.A.E.-R.; writing—original draft, G.M.-B.; writing—review and editing, J.L.V.-G., A.G.-A., A.d.J.R.-D., and F.A.M.-Z. All authors have read and agreed to the published version of the manuscript.

**Funding:** This research received no external funding.

**Acknowledgments:** The authors thank the Engineering Division, Chemical Engineering and Metallurgy Department, and the Geology Department of the University of Sonora, LANGEM, the National Laboratory of Geochemistry and Mineralogy in México, and CONACYT (National Council of Science and Technology) for the graduate scholarship of one author (G.M.-B.) and, Ing. Roberto Valenzuela Retroworks de México for their support of this study. The authors also thank Belem González Grijalva for analytical advice.

**Conflicts of Interest:** The authors declare no conflict of interest. The funders had no role in the design of the study; in the collection, analyses, or interpretation of data; in the writing of the manuscript; or in the decision to publish the results.

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
