# Peer review of "Base Metals Extraction from Printed Circuit Boards by Pressure Acid Leaching"

_minerals, doi:10.3390/min13010098_

Round 1

Reviewer 1 Report

In the new version of the manuscript, my previous comments have been taken into account.

Author Response

Thank you for your comments and recommendations to improve this article

Reviewer 2 Report

§  The language in the paper must be revised in terms of grammatical and typological errors to meet the journal's international standard.

§  The core area of this research article is leaching and separation, but how the efficiency of leaching is not considered for the other metals. Also, it is better to explain the effect of the lack or presence of other metals because A large number of base metals, such as tin, lead, cadmium, etc. can be found in waste printed boards.

§  It is suggested to use the following articles to improve the quality of your research:

Deveci, H., Aydın, U., &Akcil, A. U. 2010. Extraction of copper from scrap TV boards by sulphuric acid leaching under oxidising conditions. In Proceedings of Going Green-CARE INNOVATION 2010 Conference.

Dhawan, N., Kumar, M., Kumar, V., &Wadhwa, M. 2008. Recovery of metals from electronic scrap by hydrometallurgical route. In Proceedings of the Global Symposium on Recycling, Waste Treatment and Clean Technology (REWAS), pp. 12-15.

Gibson, R. W., Fray, D. J., Sunderland, J. G., &Dalrymple, I. M. 2003. Recovery of solder and electronic components from printed circuit boards. Electrochem. Soc. P, 18, pp. 346-354.

Kamberovic, Z. J. 2009. Hydrometallurgical process for extraction of metals from electronic waste-part I: Material characterization and process option selection.Association of Metallurgical Engineers of Serbia, 15(4), 231-243.

Lee, C. H., Chang, C. T., Fan, K. S., & Chang, T. C. 2004.An overview of recycling and treatment of scrap computers. Journal of hazardous materials, 114(1), 93-100.

M Kavousi, A Sattari, EK Alamdari, S Firozi, Selective separation of copper over solder alloy from waste printed circuit boards leach solution, Waste Management 60, 636-642

M Kavousi, A Sattari, EK Alamdari, DH Fatmehsari, Leaching studies for copper and solder alloy recovery from shredded particles of waste printed circuit boards, Metallurgical and Materials Transactions B 49 (3), 1464-1470

Yazdan Nosratzad, Maryam Kavousi, Anahita Sattari, Eskandar Keshavarz Alamdari, Davoud Haghshenas, Dariush Darvishi, Ata Keshavarz Alamdari, Ali Bagheri Kafash Rafsanjani, Recovery of Valuable Metals from E-Waste, Part (I): Use of Central Composite Design for evaluation of Copper Recovery from Electronic Waste Leaching, 24th International Mining Congress and Exhibition of Turkey- 2015

§  The condition of the experiment is not explained completely. Also, the Optimization of leaching parameters is lacking in this research article.

§  The clarity of the captions of the figures must be revised and modified. The test conditions should be written completely.

§  The reason for some fluctuation in recovery rate needs to be properly explained carefully.

§  In other studies, the sulphuric acid was used as a leaching agent for the dissolution of metals from the printed circuit boards that were found that copper and zinc leached out within 8 h by employing 2M H2SO4 and 0.2M H2O2 at 85°C, meanwhile, 95% of the iron, nickel, and aluminum were dissolved within 12 h. It seems that the results in this paper are not different from those results. (Oh CJ, Lee SO, Yang HS, Ha TJ, Kim MJ (2003) Selective leaching of valuable metals from waste printed circuit boards. J Air Waste Manag Assoc 53: 897-902.) or (High-Pressure Oxidative Leaching and Iodide Leaching Followed by Selective Precipitation for Recovery of Base and Precious Metals from Waste Printed Circuit Boards Ash, Metals - Open Access Metallurgy Journal 9(3):363) 

Reviewer 3 Report

1. Comment: Line 21. The first letter of sentences should be capitalized.

2. Comment: The logic of the introduction is not clear. The authors should clearly state the strengths of the content and methodology of this study. Highlight the significance of this study and the advantages of pressurized acid leaching.

3. Comment: Lines 168, 180 and 187. Use the same name for the Eh-pH diagram. Are the authors suggesting that Pourbaix diagram = Species diagram?

4. Comment: What is the significance of SEM-EDS analysis of PCBs before and after leaching?

5. Comment: Lines 209 and 210. "The results obtained from the EDS show mainly the presence of lead and gold." Why it is not consistent with what is being discussed. "Once the material has been leached with sulfuric acid, a morphology characterized by plates containing Ag and Au can be seen."

There is no discussion about Ag in the PCB material study of SEM-EDS leaching test, why is there a result about Ag in the discussion?

6. Comment: The scanning electron microscope (SEM-EDS) image is not clear, such as scales.

7. Comment: In this study, a large amount of spent acid was generated after leaching base metals. How to balance the relationship between environment and economic performance?

Round 2

Reviewer 2 Report

Based on the explanations sent by the authors, it seems that the comments provided have been included and the article is approved. Therefore, this article is accepted for publication.